# Is amblyopia associated with school readiness and cognitive performance during early schooling? Findings from the Millennium Cohort Study

Lisanne Andra Gitsels[1,2], Mario Cortina-Borja[1], Jugnoo Sangeeta Rahi[1,2,3,4]*

1 Population, Policy and Practice Research and Teaching Department, Great Ormond Street Institute of Child Health, University College London, London, United Kingdom, 2 Ulverscroft Vision Research Group, Great Ormond Street Institute of Child Health, University College London, London, United Kingdom, 3 Great Ormond Street Hospital for Children NHS Foundation Trust, London, United Kingdom, 4 Institute of Ophthalmology, University College London, London, United Kingdom

* j.rahi@ucl.ac.uk

**Data Availability Statement:** Data are freely available from the UK Data Archive, https://beta.ukdataservice.ac.uk/datacatalogue/series/series?id=2000031#!/access-data. For this study, we

## Abstract

### Background

Amblyopia is a neurodevelopmental condition causing reduced vision, for which pro-grammes of whole population child vision screening exist throughout the world. There is an ongoing debate about the value of screening due to the lack of evidence about meaningful functional impacts of amblyopia. Our objective was to determine whether amblyopia is asso-ciated with school readiness and early cognitive performance.

### Methods and findings

Data from the prospective Millennium Cohort Study of children born in the United Kingdom in 2000–01 and followed-up to age 7 years ($n$ = 13,967). Using parental self-report on eye conditions and treatment coded by clinical reviewers, participants were grouped into no eye conditions, strabismus alone, refractive amblyopia, or strabismic/mixed (refractive plus stra-bismic) amblyopia. The outcomes were poor school readiness using Bracken School Readi-ness Assessment <25th percentile (age 3); and cognitive tests and their age-related trajectories using British Ability Scale II Naming Vocabulary (ages 3/5) and Pattern Con-struction (ages 5/7). Multivariable analyses showed that compared to children without any eye conditions, only those with strabismic/mixed amblyopia had an increased risk of poor school readiness (OR = 2.04, 95%CI 1.09–3.82). Small differences in mean scores for NV and PC of children with amblyopia (all types) compared to those without any eye condition were not clinically significant (>10 points) irrespective of whether treatment had already started. The age-related cognitive trajectories of children with amblyopia did not differ from those without any eye conditions for either NV ($p$ = 0.62) or PC ($p$ = 0.51). These associa-tions are at population rather than individual level, so it might be that some individuals with amblyopia did experience significant adverse outcomes that are not captured by summary statistics.

made use of the first four surveys (MCS1-4 SN: 4683, 5350, 5795, and 6411). We had special access privileges as co-investigators on the CLOSER grant to the original parental report on eye conditions (variable EYEX in MCS2-4). Access is otherwise obtained via https://www.closer.ac.uk/study/millennium-cohort-study/. Information on eye conditions was included in the coding of longstanding illness (variable CLSI in MCS2-4) that is present in the freely available survey data from the UK Data Archive. The longstanding illness is based on the International Statistical Classification of Diseases and Related Health Problems 10th version (ICD-10).

**Funding:** JSR is supported by the National Institute for Health Research (NIHR) Biomedical Research Centre at Moorfields Eye Hospital NHS Foundation Trust and UCL Institute of Ophthalmology. All research at Great Ormond Street Hospital NHS Foundation Trust and UCL Great Ormond Street Institute of Child Health is made possible by the NIHR Great Ormond Street Hospital Biomedical Research Centre. The views expressed are those of the author(s) and not necessarily those of the NHS, the NIHR or the Department of Health. LAG is supported by the Ulverscroft Vision Research Group.

**Competing interests:** The authors have declared that no competing interests exist.

## Conclusions

Amblyopia is not significantly associated with adverse cognitive performance and trajectories in early schooling and there is no evidence that this is due to a mediating effect of treatment. Although amblyopia combined with strabismus is associated with poor school readiness, this is not translated into poor cognitive performance. These novel findings may explain the lack of association reported between amblyopia and educational outcomes in adult life and suggest that the impact of amblyopia on education is not of itself a justification for whole population child vision screening aimed at detecting this disorder.

## Introduction

Amblyopia (lazy eye) is a condition that has long intrigued scientists and clinicians, as both a fundamental paradigm of human neural plasticity [1] and the most common condition managed in paediatric ophthalmology [2]. Up to 5% of children in industrialised countries are affected [2, 3]. Amblyopia manifests as impaired vision, usually affecting one eye, and commonly with secondary reduction in stereo ('3D') vision. The most common predisposing disorders comprise unequal refractive status in the two eyes and strabismus (misalignment of the eyes), although both conditions can exist independently without resulting amblyopia [3]. Thus, amblyopia is commonly classified as being refractive, strabismic or mixed.

There are sensitive periods in infancy and early childhood when amblyopia develops if predisposing conditions are present. There is a time-limited critical period during childhood during which treatment can be undertaken; the upper age boundary was conventionally considered to be around age seven years but there is currently interest in the potential for novel treatments into late adolescence [4].

Amblyopia is inherently asymptomatic as a developmental disorder affecting one eye. Most industrialised countries have established population child vision screening programmes to ensure that children with amblyopia are detected early and referred promptly for treatment [3, 5]. Within the UK, screening is undertaken between the ages of four and five years in accordance with the recommendations of Public Health England's National Screening Committee [6]. Whilst screening programmes exist internationally, there remains debate about their value. This is largely due to the incomplete evidence-base about the 'real-life consequences' of amblyopia on health, well-being and social outcomes, and the extent to which these impacts are mitigated by treatment [3, 7].

We hypothesised that as reduced vision in amblyopia (all types) should impact on visual information processing and visuomotor functions, there could be discernible associations with school readiness and performance on cognitive tests that rely on these processes during the period of early childhood when amblyopia becomes manifest and treatment is undertaken. We report here an investigation of this question utilising the UK Millennium Cohort Study (MCS) [8].

## Methods

### Study design

We analysed data from the MCS, a prospective population-based cohort study of children born in the UK in 2000–01 that used a stratified sampling design to achieve an overrepresentation of families from an ethnic minority group and deprived background [8]. This makes the

cohort particularly suitable for research on amblyopia, which varies by ethnicity and socioeconomic status [2]. Drawing on our prior work on children with eye conditions in the MCS at age three years [9, 10] we used the subsequent follow-up data to 7 years, an age at which all children with amblyopia would be expected to be detected and treatment largely completed.

## Participants

In the absence of formal clinical assessments, parents reported on their child's vision and eye conditions at ages 3, 5, and 7 years using open and closed-ended questions [11]. The responses were coded based on a taxonomy applied in our previous research on childhood blindness [12] drawing on the International Classifications of Diseases (ICD). This extended coding has been successfully validated and subsequently used [9, 10]. Coding was undertaken independently by three researchers (PC, MCIB and LAG) and discordant codes were verified by a consultant ophthalmologist (JSR). A conservative approach was used whereby at least one parental report of both eye condition(s) and treatment(s) at ages 3, 5, or 7 had to fully match. The treatments were surgery, occlusion by patch or penalisation using cycloplegic drops, and spectacles.

In order to understand the impact on cognitive abilities and trajectories of visual information processing, we categorised children into four mutually exclusive groups comprising two affected groups of a) refractive amblyopia and b) strabismic or mixed (strabismic and refractive) amblyopia along with two comparator groups of c) no eye conditions (main reference) and d) strabismus alone, the latter to differentiate between the impact of impaired acuity (amblyopia) and ocular misalignment *per se*. Refractive amblyopia has the best prognosis among amblyopia types, affecting mainly acuity, whilst strabismic amblyopia directly impact additionally on stereovision, such that mixed amblyopia is clinically the most 'severe' amblyopia. Furthermore, delineation of refractive and strabismic or mixed amblyopia distinguishes between children more likely to be diagnosed later e.g. at screening, or earlier because they had cosmetically obvious strabismus.

Children with any other eye conditions, neurological or -developmental conditions (such as cerebral palsy and Down syndrome) were excluded as these conditions could themselves impact on both cognitive outcomes and vision (S1 Table) [9, 10, 13]. Children from multiple births were excluded as they are more likely to have vision disorders, like strabismus [10].

## Outcomes

We were interested in school readiness and cognitive performance in early childhood, as both are strongly associated with educational attainment, income, and health later in life [14–16]. In the MCS, school readiness is measured by the Bracken School Readiness Assessment Revised (BSRA-R) at age 3 [14] and cognitive abilities and their age-related trajectories are assessed by the British Ability Scale (BAS) II Naming Vocabulary at ages 3 and 5 and BAS II Pattern Construction at ages 5 and 7 [15]. The composite of the BSRA-R consists of six subtests that assess educationally relevant concepts needed for early formal education [14]. These subtests comprise knowledge of colours, letters, shapes, numbers and counting and the child's ability to describe, match and compare objects. Poor school readiness was defined as a score in or below the 25th percentile. The BAS II is a standardised cognitive test battery in the UK to evaluate the developmental and learning abilities of children [15]. Naming Vocabulary is picture-based and assesses expressive language ability and long-term memory as part of the visual-verbal information processing [15]. Pattern Construction is object-based and assesses visual perception and spatial awareness as part of the visual-spatial information processing [15]. Children are expected to have higher scores with increasing age, reflecting maturing of

cognitive performance and greater educational experience. The scores range from 20 to 80 points with a mean of 50. Clinically significant thresholds are differences greater than 10 or 12 points in Naming Vocabulary at age 3 or 5, and 12 or 14 points in Pattern Construction at age 5 or 7 [15].

## Statistical analyses

Potential confounders in the association of amblyopia and educational outcomes comprised child's sex, ethnicity, gestational age, and birth order, maternal education, and household's main language and disposable income [2, 10, 13, 17–19], measured at baseline age 9 months (S1 Table).

All analyses used survey weights to adjust for the survey design and attrition over time. Missingness in the baseline characteristics was dealt with by listwise deletion, multiple imputations, or exclusion of variable if its proportion was <5%, 5–49%, or ≥50%, respectively [20]. Missingness in the outcome variables was dealt with by pairwise deletion to maximise the sample size for each outcome of interest. Patterns of missing data, found by logistic regressions, may bias the generalisability of the results towards the group which is overrepresented in the complete sample.

Chi-squared tests assessed proportion differences in amblyopia and/or strabismus status by baseline characteristics and age at which treatment had started. Among children with amblyopia and/or strabismus, logistic regression models of treatment started by age 3 or 5 assessed associations with type of eye condition and baseline characteristics. A logistic regression model of school readiness and two linear mixed effects (LME) regression models of Naming Vocabulary and Pattern Construction were fitted to assess associations with amblyopia and/or strabismus, adjusted for all baseline characteristics as fixed effects and child level as a random effect to account for the repeated measures of cognitive tests. Interactions between amblyopia and/or strabismus, sex and age were only included if significant at the 5% significance level. The model assumptions were checked (S1 File). MCS data were obtained from the UK Data Archive, University of Essex (Essex, England), and analysed in R version 3.5.3 [21] using packages `survey` and `lme4`.

## Ethics and patient and public involvement

The MCS was approved by the relevant Ethics Committees [8] and families of participants gave informed consent to participate [8]. Although the MCS is active in participant and public engagement, there was no direct involvement in this specific study, which drew on existing data.

## Results

### Study population

The analysis sample consisted of 13,967 singletons aged 3 years old after exclusions due to selection criteria and missingness (Fig 1). Of these 9835 (68%) completed all cognitive tests with the sample size for each outcome shown in Fig 1. Missing outcomes were not associated with amblyopia and/or strabismus status, however white, English-speaking, higher qualified, and richer families are moderately overrepresented in the complete-data samples (S2 Table).

The proportion of children with any amblyopia overall was 2.2% (2.0 to 2.5)–mostly unilateral (97%)–and 2.6% (2.2 to 2.9) children had strabismus (Table 1). The proportions varied by ethnicity, gestational age, and household income (Table 1), in keeping with the current literature [3]. Data are not available on whether participants had undergone child vision screening.

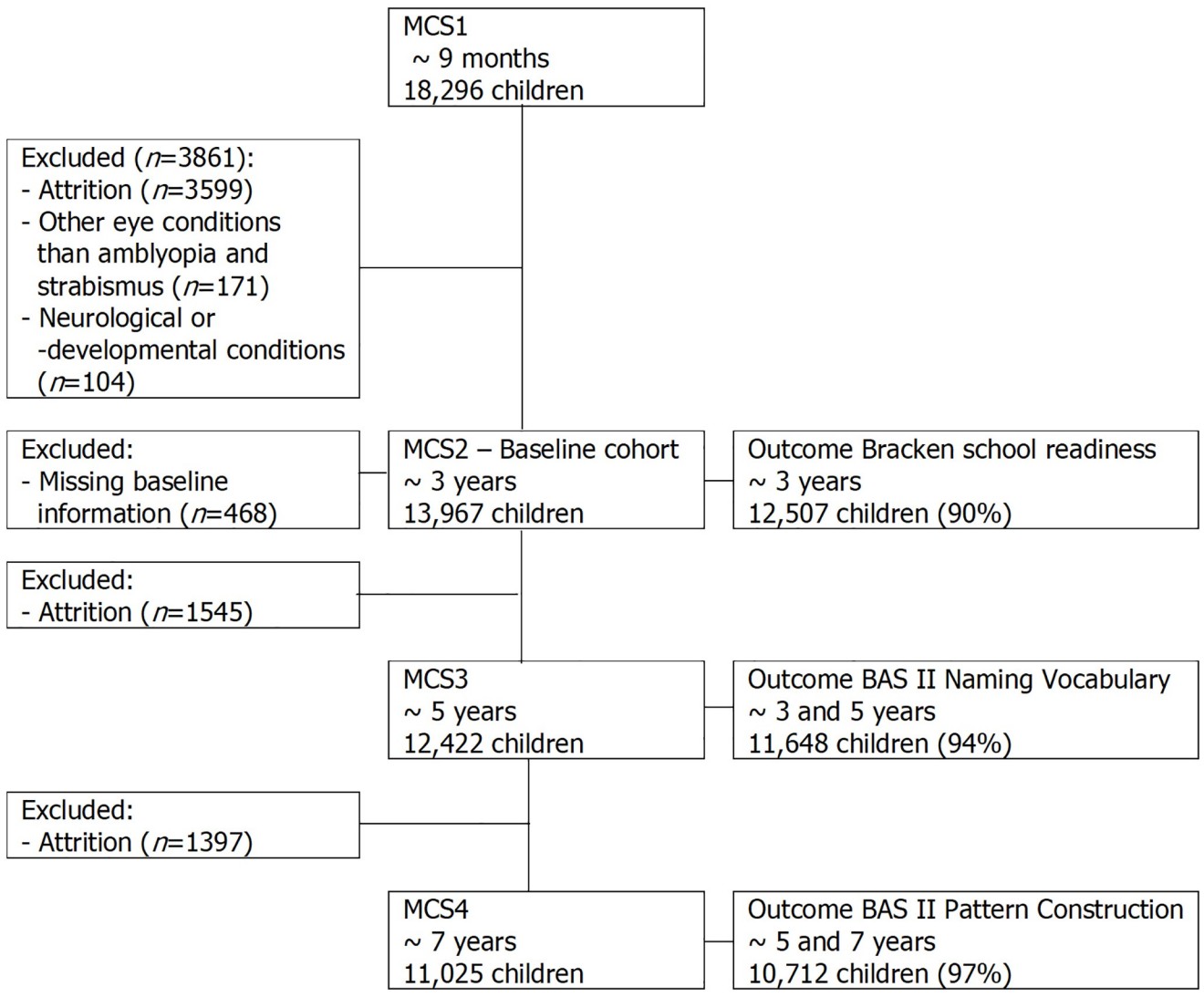

**Fig 1. Study population of singletons.**

However, there were significant differences (*p*<0.001) in the age of starting treatment by amblyopia and/or strabismus status with all having started by age 7 (Fig 2). Among children with amblyopia and/or strabismus, the proportions of treatment started by age 3 and 5 did not vary by sex, gestational age, birth order, maternal education, and household language and income when adjusted for type of eye condition (Table 2). However, children from any non-white ethnic group were less likely to had started treatment by age 5 compared to white children (Adjusted Odds Ratio (AOR) = 0.41 (0.17 to 0.99)).

## School readiness

Only children with strabismic or mixed amblyopia were at greater risk of poor school readiness at age 3 compared to children without any eye conditions (AOR = 2.04 (1.09 to 3.82)) (Table 3). This effect size was comparable to the associations of poor school readiness with non-white ethnicity, higher birth order, lower maternal educational level, English not as main

**Table 1. Baseline characteristics of study population at age 3 years by ever presence amblyopia and/or strabismus status.**

| Covariate | Category | No eye condition, $n$ = 13,431 (weighted %) | Strabismus alone, $n$ = 264 (weighted %) | Refractive amblyopia, $n$ = 188 (weighted %) | Strabismic or mixed amblyopia, $n$ = 84 (weighted %) | $\chi^2$ global (weighted) $p$-value |
|---|---|---|---|---|---|---|
| **Sex** | Girls | 6568 (49%) | 131 (49%) | 95 (51%) | 34 (42%) | 0.65 |
| | Boys | 6863 (51%) | 133 (51%) | 93 (49%) | 50 (58%) | |
| **Ethnicity** | White | 11535 (89%) | 240 (94%) | 172 (95%) | 77 (92%) | <0.01 |
| | Non-white | 1896 (11%) | 24 (6%) | 16 (5%) | 7 (8%) | |
| **Birth order** | 1 | 6814 (51%) | 155 (61%) | 102 (54%) | 45 (55%) | 0.10 |
| | 2 | 4056 (31%) | 67 (27%) | 46 (26%) | 24 (25%) | |
| | 3+ | 2561 (18%) | 42 (12%) | 40 (20%) | 15 (20%) | |
| **Gestational age** | ≥37 weeks | 12,565 (94%) | 230 (84%) | 173 (91%) | 74 (91%) | <0.01 |
| | <37 weeks | 866 (6%) | 34 (16%) | 15 (9%) | 10 (9%) | |
| **Maternal education** | A-levels or higher | 4876 (37%) | 78 (33%) | 63 (33%) | 21 (27%) | 0.30 |
| | O-levels | 6265 (48%) | 131 (49%) | 87 (47%) | 42 (55%) | |
| | None | 2290 (15%) | 55 (18%) | 38 (21%) | 21 (18%) | |
| **Household language** | English | 12989 (94%) | a | a | a | NA |
| | Non-English | 422 (6%) | a | a | a | |
| **Household income** | ≥£20800 | 4858 (39%) | 72 (31%) | 57 (32%) | 17 (25%) | 0.02 |
| | £10400-£20800 | 4392 (32%) | 89 (32%) | 61 (33%) | 32 (40%) | |
| | <£10400 | 4181 (29%) | 103 (37%) | 70 (36%) | 35 (35%) | |
| **Total** | Proportion (95%CI) | 95.9% (95.5–96.3) | 1.9% (1.6–2.2) | 1.5% (1.3–1.8) | 0.7% (0.6–0.9) | |

[a] Not provided to avoid potential disclosure.

household language, and lower household income. There was no association between refractive amblyopia or strabismus alone and school readiness nor was there an association with starting treatment by age three years. There was no significant interaction between amblyopia and/or strabismus status and sex associated with school readiness ($p$ = 0.40).

## Cognitive abilities and trajectories

After adjusting for all covariates, children without any eye conditions had a mean of 57.6 (56.9 to 58.2) points on Naming Vocabulary at age 3 (Fig 3, S3 Table). By comparison, children with strabismic or mixed amblyopia and with refractive amblyopia had an adjusted mean of 3.2 (1.0 to 5.4) and 1.3 (-0.1 to 2.7) points lower, respectively, whilst those with strabismus alone had 0.8 (-0.8 to 2.3) points lower. Children without any eye conditions had a mean of 53.4 (52.5 to 54.3) points on Pattern Construction at age 5. By comparison, children with strabismic or mixed amblyopia and with strabismus alone had an adjusted mean of 2.7 (0.7 to 4.8) and 2.7 (0.1 to 5.3) points lower, whereas those with refractive amblyopia had 1.1 (-0.9 to 3.0) points lower. None of these was significantly different at clinical thresholds. Furthermore, age at which treatment had started was not associated with Naming Vocabulary ($p$ = 0.43) and Pattern Construction ($p$ = 0.14). As expected, children scored higher on the cognitive tests at older ages; children without any eye conditions had a mean of 4.1 (3.9 to 4.2) points higher in Naming Vocabulary at age 5 compared to age 3, and 1.8 (1.6 to 1.9) points higher in Pattern Construction at age 7 compared to age 5. Notably, children with amblyopia and/or strabismus

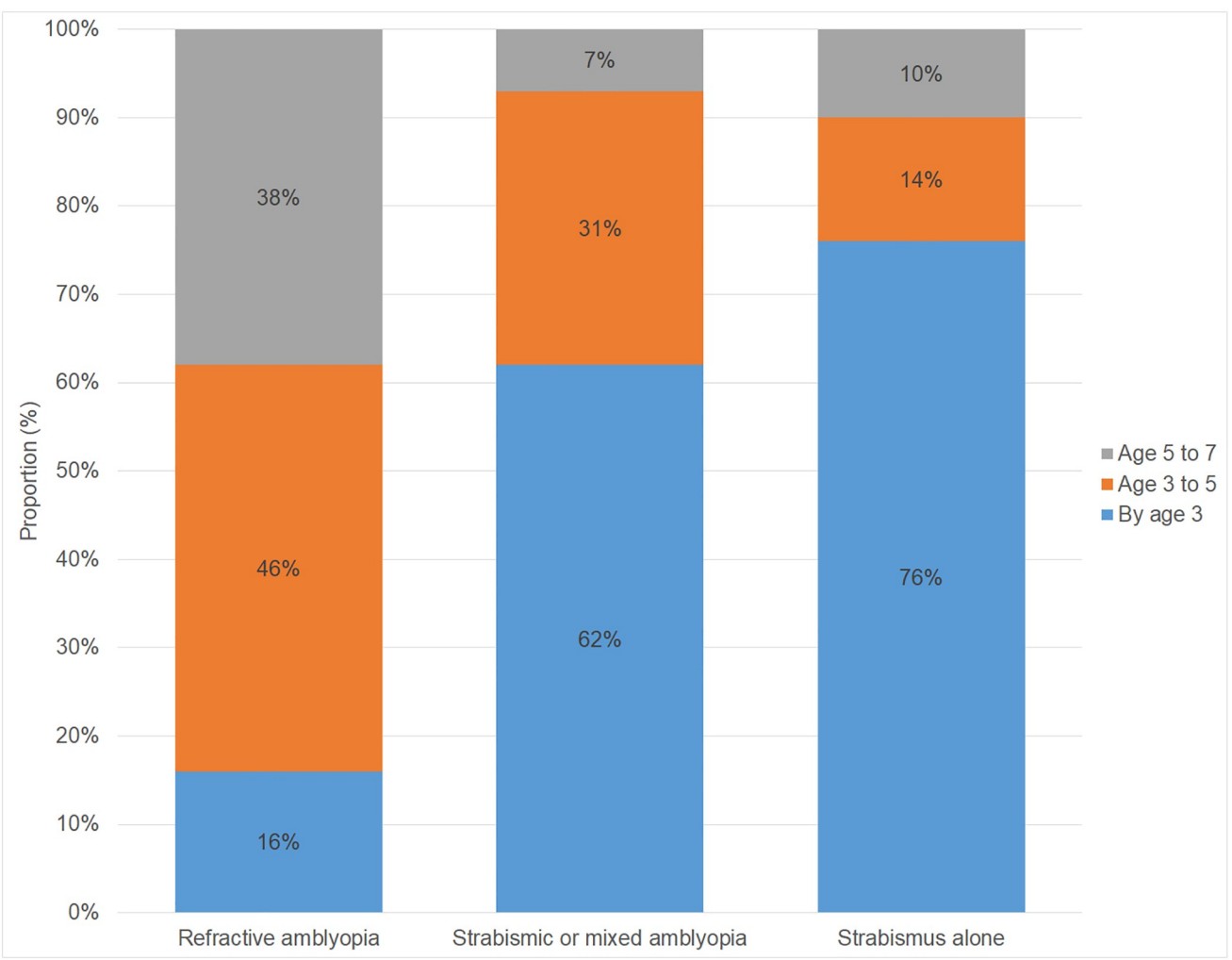

**Fig 2. Age of starting treatment by amblyopia and/or strabismus status.**

had similar cognitive trajectories of Naming Vocabulary ($p = 0.62$) and Pattern Construction ($p = 0.51$) to those without any eye conditions.

Cognitive trajectories (shown in Fig 3) differed by sex ($p<0.001$), where boys scored on average lower than girls at younger ages but made greater progress over time. There was no significant interaction between amblyopia and/or strabismus status and sex associated with cognitive trajectories in Naming Vocabulary ($p = 0.28$) and Pattern Construction ($p = 0.26$). These cognitive tests were associated with ethnicity, birth order, gestational age, maternal education, and household language and income, except for no associations of Pattern Construction with birth order and household language. None of these differences, however, were significantly different at clinical thresholds.

## Discussion

From a population-based cohort study, no differences were found between children with amblyopia (any type) and those without any eye conditions or those with strabismus alone in cognitive performance and trajectories between the ages 3 and 7 years, i.e. during the period in

**Table 2. Odds of treatment started by age 3 and 5 among children with amblyopia and/or strabismus.**

| Covariate | Category | Age 3 Adjusted[a] OR (95%CI) | Age 5 Adjusted[a] OR (95%CI) |
|---|---|---|---|
| **Eye condition** | Strabismus alone | 1.00 | 1.00 |
| | Refractive amblyopia | 0.07 (0.04–0.12) | 0.21 (0.12–0.38) |
| | Strabismic/mixed amblyopia | 0.54 (0.29–0.99) | 2.16 (0.71–6.57) |
| **Sex** | Girls | 1.00 | 1.00 |
| | Boys | 1.26 (0.79–2.03) | 0.74 (0.43–1.27) |
| **Ethnicity** | White | 1.00 | 1.00 |
| | Non-white | 0.94 (0.42–2.08) | 0.41 (0.17–0.99) |
| **Birth order** | 1 | 1.00 | 1.00 |
| | 2 | 1.03 (0.59–1.81) | 0.78 (0.41–1.49) |
| | 3+ | 1.02 (0.53–1.96) | 0.94 (0.46–1.95) |
| **Gestational age** | ≥37 weeks | 1.00 | 1.00 |
| | <37 weeks | 1.10 (0.47–2.57) | 0.69 (0.28–1.71) |
| **Maternal education** | A-levels or higher | 1.00 | 1.00 |
| | O-levels | 0.99 (0.56–1.73) | 1.12 (0.59–2.12) |
| | None | 1.38 (0.65–2.97) | 1.22 (0.52–2.87) |
| **Household language** | English | 1.00 | 1.00 |
| | Non-English | 0.54 (0.08–3.47) | 1.93 (0.31–11.91) |
| **Household income** | ≥£20800 | 1.00 | 1.00 |
| | £10400-£20800 | 1.08 (0.58–2.01) | 1.14 (0.56–2.34) |
| | <£10400 | 0.96 (0.52–1.77) | 0.98 (0.48–1.99) |

[a] Odds ratios adjusted for all covariates and sample weights.

which amblyopia would be detected and treated. Strabismic or mixed amblyopia, but not refractive amblyopia or strabismus alone, was associated with poor school readiness at age 3 years. Notably, there were no associations between treatment and school readiness nor cognitive performance and trajectories.

The MCS had particular advantages for this investigation because of its overall size and over-sampling of subgroups known to be at greater risk of both amblyopia and strabismus [2] and poor educational outcomes [22]. Standardised, age appropriate measures of cognitive ability that require visual information processing were taken at multiple ages in the same children, enabling, for the first time, age-related trajectories to be investigated during the period in which amblyopia and strabismus are detected. Exclusion of children with any other eye conditions as well as those with neurological or -developmental disorders ensured investigation specifically of the impact of amblyopia.

Although a limitation of the study was reliance on parental self-report for the classification of eye conditions, the interview questions were carefully constructed at the outset by ophthalmologists and all coding was undertaken by us and has been validated [9, 10]. As the frequencies of both amblyopia and strabismus in MCS align closely with similar population studies in the UK [2] and the pattern of age at starting treatment by type of amblyopia aligns with clinical expectations, there is unlikely to be meaningful misclassification bias of eye conditions.

The MCS, like most longitudinal studies, had attrition which was addressed by using survey weights in the analyses. Missing data in the baseline characteristics were negligible and not associated with the outcomes, but missing data in the outcomes reduced the sample size by up to 10%. Despite adjustment of analyses for known confounders, the possibility of residual confounding remains. Finally, as with any epidemiological study, associations are reported at the

**Table 3. Odds of school readiness in <25th percentile at age 3.**

| Covariate | Category | Adjusted[a] OR (95%CI) |
|---|---|---|
| **Eye condition** | No eye condition | 1.00 |
| | Strabismus alone | 1.02 (0.69–1.51) |
| | Refractive amblyopia | 1.04 (0.68–1.60) |
| | Strabismic/mixed amblyopia | 2.04 (1.09–3.82) |
| **Treatment** | Started by age 3 | 1.00 |
| | Not started by age 3 | 1.11 (0.86–1.43) |
| **Sex** | Girls | 1.00 |
| | Boys | 1.78 (1.63–1.94) |
| **Ethnicity** | White | 1.00 |
| | Non-white | 2.06 (1.75–2.43) |
| **Birth order** | 1 | 1.00 |
| | 2 | 1.19 (1.08–1.32) |
| | 3+ | 2.03 (1.82–2.28) |
| **Gestational age** | ≥37 weeks | 1.00 |
| | <37 weeks | 1.30 (1.01–1.53) |
| **Maternal education** | A-levels or higher | 1.00 |
| | O-levels | 1.96 (1.73–2.22) |
| | None | 3.07 (2.60–3.62) |
| **Household language** | English | 1.00 |
| | Non-English | 1.73 (1.23–2.43) |
| **Household income** | ≥£20800 | 1.00 |
| | £10400-£20800 | 1.70 (1.52–1.91) |
| | <£10400 | 3.42 (3.04–3.85) |

[a] Odds ratios adjusted for all covariates and sample weights.

population rather than individual level, so it might be that some individuals with amblyopia did experience significant adverse outcomes that are not captured by summary statistics.

The ongoing debate about the value of whole population vision screening to detect amblyopia centres largely on the question of whether there are meaningful functional impacts of amblyopia and the extent to which these can be mitigated by treatment in early childhood. Children with amblyopia typically develop visual deficit early in childhood and thus never experience 'normal' vision in the affected eye, which explains the 'asymptomatic' nature of amblyopia and the need for screening to detect affected individuals. Thus, the extensive evidence-base delineating the functional impact of bilateral visual impairment on physical, social and emotional development during childhood and wide-ranging health, social and economic outcomes throughout life is not germane to the current study. Nor is the research on the impact of visual loss acquired later in life [23, 24].

There is some evidence of deficits in the performance of specific visuomotor or -cognitive tasks by children with amblyopia, for example, discernibly impaired of visuomotor tasks if there is also reduced stereovision [25]. This might explain our finding that on average children with strabismus (with/without amblyopia) scored the lowest on Pattern Construction. Equally, although amblyopic children aged 8 to 12 years have been reported to have slower reading speed than peers without eye conditions [26, 27], reading abilities and the trajectory of reading

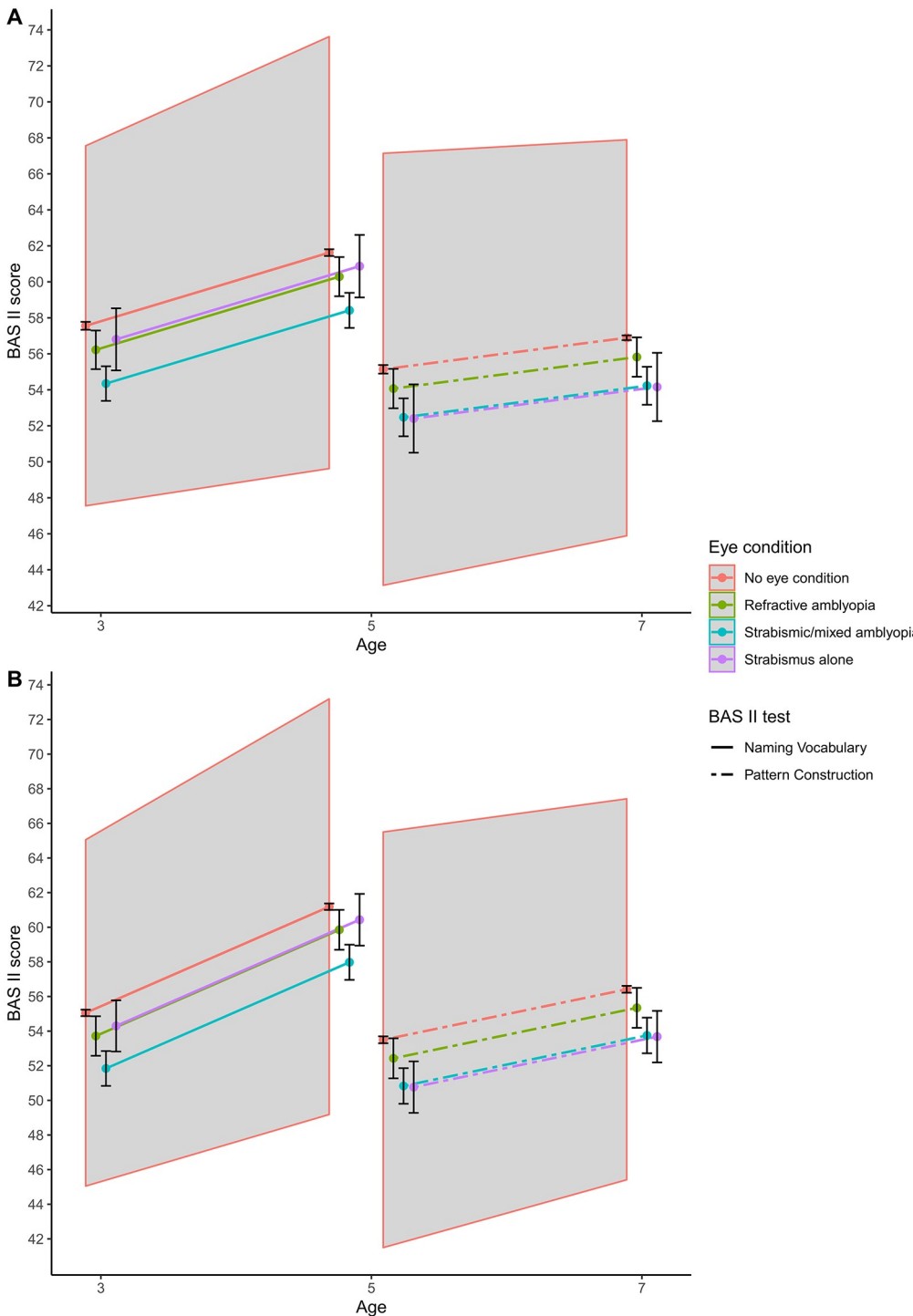

**Fig 3. Age-related cognitive trajectories using British Ability Scale II tests by eye condition in (A) girls and (B) boys.** Scores for baseline children of white ethnicity, gestational age ≥37 weeks, firstborn, whose mothers had A-levels or higher qualification, had English as main household language and were from a median or richer household. Scores were adjusted for sample weights and a random effect at child level. The shaded area is the age-related non-significant clinical difference from the reference group of no eye condition. Age at starting treatment was not associated with Naming Vocabulary ($p = 0.43$) and Pattern Construction ($p = 0.14$).

development of amblyopic adolescents are reported not to differ meaningfully from those without eye conditions [28]. There is very little evidence that these specific deficits in vision-dependent tasks in amblyopes translate into poorer performance, abilities or outcomes in meaningful and important 'real-life' activities or tasks [3, 29].

There are few studies with which we can compare our findings directly. However, studies reporting on educational outcomes are indirectly relevant, as they speak to the combination of cognitive ability in early childhood and learning experience during schooling/higher education [15]. An investigation of the ALSPAC cohort in one region of England reported that reduced acuity in one eye (including but not restricted to amblyopia) was not associated with lower attainment in standardised assessments of English, Maths, and Science at ages 10–11 and 14–16, respectively [13]. Our own investigation of the 1958 British Birth Cohort showed that amblyopia was not associated with educational test performances at age 11 nor educational attainment at age 33 [17]. Amblyopia did not affect lifetime occupational class but fewer affected individuals completed a university degree in an Australian study [30]. Our finding of a lack of association of either amblyopia (all types) or strabismus alone on cognitive abilities requiring visual-verbal and visual-spatial processing and their trajectories from age 3 to 5 to 7 is novel. This may provide insights into the absence of evidence of differences between those with and without amblyopia in terms of visuospatial cognition [31] and educational outcomes [13, 28, 30, 32] later in life.

Whole population child vision screening was well established in the UK at the time that participants in the MCS would have been eligible, although whether screening was undertaken was not recorded. One rationale for the recommendation from the UK National Screening Committee that screening should be undertaken at school entry (versus prior standalone pre-school screening programmes) is to ensure equitable provision and access in the face of evidence of socio-economic variations in uptake [2, 33]. Our findings that the likelihood of starting treatment for amblyopia was not associated with gender, gestational age, maternal education, or household income, provide indirect support that the currently recommended universal child vision programme in the UK is successfully addressing some prior inequalities. Nevertheless, the association between non-white ethnicity and reduced odds of starting treatment by age 5 years is noteworthy and requires further attention. Our finding that most children with strabismus (with/without amblyopia) had started treatment by age 3 whereas those with refractive amblyopia at older ages, supports the notion that screening at age 4–5 years does indeed—as intended—identify those with asymptomatic impaired acuity rather than those with cosmetically obvious strabismus. Whilst our ability to understand the mediating role of treatment is limited due to the nature of the MCS dataset, the lack of any associations between having started treatment and school readiness or cognitive performance is notable. There is no reason to think these findings are not generalisable to other similar populations.

Children with amblyopia combined with strabismus were found to be at meaningfully increased risk of poor school readiness at age 3 years and it is notable that the size of this association was comparable to key factors that are targeted in interventions to improve school readiness [16]. The mechanism underlying this association is not known but as there was no association with treatment this suggests that it is not mediated by visual function. There is merit in research to understand whether amblyopia combined with strabismus is a marker of broader vulnerability that manifests as poorer school readiness.

Our findings are consistent with the notion that amblyopia *per se* does not have a meaningful functional impact on cognitive abilities in early childhood and this lack of association is not attributable to treatment. Taken together with existing evidence of no associations with educational outcomes later in life, these findings suggest that averting adverse educational experience in attainment is not a reasonable justification of itself for screening to detect amblyopia.

## Supporting information

**S1 Table. Coding of covariates.**
(DOCX)

**S2 Table. Odds of missing data in confounders and cognitive tests.** [a] Missing data in confounders (n = 471): gestational age (n<10), maternal education (n = 32), and disposable household income (n = 445). [b] Missing data cognitive tests (based on the number of children participating at the oldest age at which the cognitive test was taken): Bracken School Readiness Assessment-Revised (n = 1460), British Ability Scale II Naming Vocabulary at ages 3 and 5 (n = 774), and British Ability Scale II Pattern Construction at ages 5 and 7 (n = 313). [c] Odds ratios adjusted for all covariates listed in the table and sample weights.
(DOCX)

**S3 Table. Associations with cognitive abilities and age-related trajectories using British Ability Scale II tests.** [a] BAS II NV, British Ability Scale II Naming Vocabulary at ages 3 and 5; BAS II PC, British Ability Scale II Pattern Construction at ages 5 and 7. [b] Estimates adjusted for all covariates, sample weights, and random effect on child.
(DOCX)

**S1 File. Regression model assumptions.**
(DOCX)

## Acknowledgments

We would like to thank Phillippa Cumberland and Maria C Ibáñez-Bruron for the coding of eye conditions.

## Disclaimer

The views expressed in this publication are those of the authors and not necessarily those of the NHS or NIHR.

## Author Contributions

**Conceptualization:** Jugnoo Sangeeta Rahi.

**Formal analysis:** Lisanne Andra Gitsels, Mario Cortina-Borja.

**Methodology:** Lisanne Andra Gitsels, Mario Cortina-Borja, Jugnoo Sangeeta Rahi.

**Visualization:** Lisanne Andra Gitsels, Mario Cortina-Borja.

**Writing – original draft:** Lisanne Andra Gitsels, Jugnoo Sangeeta Rahi.

**Writing – review & editing:** Lisanne Andra Gitsels, Mario Cortina-Borja, Jugnoo Sangeeta Rahi.

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
