## [Decision Letter · Decision Letter 0]

15 May 2020

PONE-D-20-10720

Is amblyopia associated with school readiness and cognitive performance during early schooling? Findings from the Millennium Cohort Study

PLOS ONE

Dear Professor Rahi,

Thank you for submitting your manuscript to PLOS ONE. After careful consideration, we feel that it has merit but does not fully meet PLOS ONE’s publication criteria as it currently stands. Therefore, we invite you to submit a revised version of the manuscript that addresses the points raised during the review process.

AE-1 Please address as a limitation to the study that the main variables studied such as amblyopia, and strabismus were self reported.

AE-2 Please provide some details about the Bracken School Readiness Assessment Revised (BSRA-R) and the British Ability Scale.

We would appreciate receiving your revised manuscript by Jun 29 2020 11:59PM. To enhance the reproducibility of your results, we recommend that if applicable you deposit your laboratory protocols in protocols.io, where a protocol can be assigned its own identifier (DOI) such that it can be cited independently in the future. For instructions see: http://journals.plos.org/plosone/s/submission-guidelines#loc-laboratory-protocols

We look forward to receiving your revised manuscript.

Kind regards,

Ahmed Awadein, MD, Ph.D, FRCS

Academic Editor

PLOS ONE

Journal Requirements:

Reviewers' comments:

Reviewer's Responses to Questions

**Comments to the Author**

1. Is the manuscript technically sound, and do the data support the conclusions?

Reviewer #1: Yes

Reviewer #2: Yes

2. Has the statistical analysis been performed appropriately and rigorously? 

Reviewer #1: Yes

Reviewer #2: Yes

3. Have the authors made all data underlying the findings in their manuscript fully available?

Reviewer #1: Yes

Reviewer #2: Yes

4. Is the manuscript presented in an intelligible fashion and written in standard English?

Reviewer #1: Yes

Reviewer #2: Yes

5. Review Comments to the Author

Reviewer #1: The study explored whether amblyopia is associated with school readiness and early cognitive performance based on the data from the prospective Millennium Cohort Study of children born in the United Kingdom. The paper is well written and conclusions are sound.

Questions:

Line 101. "Children with any other eye conditions were excluded". Which eye conditions were excluded and what impact it could have had on the findings of the study.

Line 154 "white, English-speaking, higher qualified, and richer families are moderately overrepresented in the complete-data samples" Why are samples not representative of the UK population. What impact it could have on the findings of the study.

Were white ethnicity, higher birth order, lower maternal educational level, English not as main household language, and lower household income associated with amplyopia?

Reviewer #2: Dear Authors, thank you for your work. I appreciate the effort of conducting and finalizing such an interesting and useful study for Pediatric Ophthalmologists. The study results fit in my long practice experience results and conclusions.

Best wishes,

Daniela Cioplean

6. PLOS authors have the option to publish the peer review history of their article (what does this mean?). If published, this will include your full peer review and any attached files.

Reviewer #1: No

Reviewer #2: Yes: Daniela Eleonora Cioplean

---

## [Author Response · Author response to Decision Letter 0]

19 May 2020

Dear Prof Awadein, 

Thank you for the expert review of our manuscript titled “Is amblyopia associated with school readiness and cognitive performance during early schooling? Findings from the Millennium Cohort Study” (manuscript ID PONE-D-20-10720). 

We appreciate your comments and that of the two external reviewers. The comments and our response after each comment in italics are provided below. A clean revised manuscript and a marked-up copy of the changes made in the original manuscript and supplementary file are submitted as part of the revision. 

Academic editor

AE-1 Please address as a limitation to the study that the main variables studied such as amblyopia, and strabismus were self reported.

Thank you for your comment. We have put more emphasis on the limitation that the classification of eye conditions was based on parental self-report, please see page 12 lines 284-286:

“Although a limitation of the study was reliance on parental self-report for the classification of eye conditions, the interview questions were carefully constructed at the outset by ophthalmologists and all coding was undertaken by us and has been validated (9,10). As the frequencies of both amblyopia and strabismus in MCS align closely with similar population studies in the UK (2) and the pattern of age at starting treatment by type of amblyopia aligns with clinical expectations, there is unlikely to be meaningful misclassification bias of eye conditions.” 

AE-2 Please provide some details about the Bracken School Readiness Assessment Revised (BSRA-R) and the British Ability Scale.

We added more information on the outcomes, please see page 5 lines 124-129: 

“The composite of the BSRA-R consists of six subtests that assess educationally relevant concepts needed for early formal education (14). These subtests comprise knowledge of colours, letters, shapes, numbers and counting and the child’s ability to describe, match and compare objects. […] The BAS II is a standardised cognitive test battery in the UK to evaluate the developmental and learning abilities of children (15).”

Reviewer #1 

The study explored whether amblyopia is associated with school readiness and early cognitive performance based on the data from the prospective Millennium Cohort Study of children born in the United Kingdom. The paper is well written and conclusions are sound.

Questions:

Line 101. "Children with any other eye conditions were excluded". Which eye conditions were excluded and what impact it could have had on the findings of the study.

Thank you for your comment. The aim of our study was to investigate the specific impact of amblyopia – which most commonly causes unilateral impairment of vision with often only mildly or moderately reduced acuity in the affected eye but generally some impairment of stereo vision. We therefore deliberately excluded all children who would have conditions that caused visual impairment or blindness (by definition disorders affecting both eyes) as we know that significantly impaired vision in both eyes is definitely associated with challenges in the educational setting. Therefore we excluded children with any visual impairment or blindness due to any cause including cerebral visual impairment and the following ocular conditions: nystagmus, ptosis, anophthalmos, glaucoma, anterior segment abnormality, craniofacial disorders, corneal opacity and dystrophy, cataract, retinal detachment, retinal dystrophy, albinism, retinopathy of prematurity, retinal haemorrhage, retinal coloboma, retinitis, coloboma, aniridia, uveal tumour, and uveitis, optic nerve hypoplasia, secondary atrophy, optic neuropathy/neuritis, optic nerve tumour, coloboma, and ocular trauma to the eye. We added this information to Table S1 in the Supplementary File and refer to it in the Manuscript on page 5 line 115. As written in the Discussion Section on page 12 lines 282-283, the exclusion of children with any other eye conditions ensured investigation specifically of the impact of amblyopia. If these children were to be included with the control group, it would have diluted the impact of amblyopia compared to normal vision associated with school readiness and cognitive abilities.

Line 154 "white, English-speaking, higher qualified, and richer families are moderately overrepresented in the complete-data samples" Why are samples not representative of the UK population. What impact it could have on the findings of the study.

All statistical analyses used survey weights to correct for the unequal probabilities of selection that resulted from the stratified cluster survey design and loss to follow-up across sweeps. However, they do not correct for partial missing information. In this study, there were missing data in the outcomes (up to 10%), resulting in some characteristics being moderately overrepresented in the complete-data samples. Ethnicity, language, maternal education and income were associated with missingness, but more importantly, amblyopia and/or strabismus status was not, therefore unbiased estimates of amblyopia and/or strabismus status associated with the outcomes could still be obtained when adjusting for the other factors. 

Were white ethnicity, higher birth order, lower maternal educational level, English not as main household language, and lower household income associated with amplyopia?

Indeed ethnicity and household income were associated with amblyopia, however birth order and maternal education were not. These results can be seen in Table 1 on page 9 and the significant ones were reported on page 7 lines 175-177.

Reviewer #2 

Dear Authors, thank you for your work. I appreciate the effort of conducting and finalizing such an interesting and useful study for Pediatric Ophthalmologists. The study results fit in my long practice experience results and conclusions.

Best wishes,

Daniela Cioplean

Thank you for the compliment.

We are grateful for the work you and the reviewers have conducted to improve our manuscript. We hope that we have addressed all the concerns raised and that our paper could be published in your journal. 

We are looking forward to hearing from you. 

Sincerely,

Jugnoo Rahi

Professor of Ophthalmic Epidemiology and Honorary Consultant Ophthalmologist

Population, Policy and Practice Research and Teaching Department

University College London Great Ormond Street Institute of Child Health

30 Guilford Street, London, WC1N 1EH, UK

---

## [Editor Report · Decision Letter 1]

27 May 2020

Is amblyopia associated with school readiness and cognitive performance during early schooling? Findings from the Millennium Cohort Study

PONE-D-20-10720R1

Dear Dr. Rahi,

We are pleased to inform you that your manuscript has been judged scientifically suitable for publication and will be formally accepted for publication once it complies with all outstanding technical requirements.

With kind regards,

Ahmed Awadein, MD, Ph.D, FRCS

Academic Editor

PLOS ONE
---

## [Editor Report · Acceptance letter]

10 Jun 2020

PONE-D-20-10720R1 

Is amblyopia associated with school readiness and cognitive performance during early schooling? Findings from the Millennium Cohort Study 

Dear Dr. Rahi:

I'm pleased to inform you that your manuscript has been deemed suitable for publication in PLOS ONE. Congratulations! Your manuscript is now with our production department. 

Kind regards, 

on behalf of

Dr. Ahmed Awadein 

Academic Editor

PLOS ONE